# Accurate Design of Microwave Filter Based on Surrogate Model-Assisted Evolutionary Algorithm

**Yongliang Zhang [1], Xiaoli Wang [2], Yanxing Wang [2], Ningchaoran Yan [2], Linping Feng [3] and Lu Zhang [1,***

1    The College of Transportation, Inner Mongolia University, Hohhot 010021, China
2    The School of Electronics and Information Engineering, Inner Mongolia University, Hohhot 010021, China
3    The School of Electronics and Information Engineering, Xi'an Jiaotong University, Xi'an 710049, China
*    Correspondence: imuzhanglu@imu.edu.cn

**Abstract:** Filter optimization problems involve time-consuming simulations and many variables in the design. These problems require a large amount of computation. This paper proposes an adaptive online updating 1D convolutional autoencoders (AOU-1D-CAE) surrogate model for solving this computationally expensive problem. In the optimization process, an adaptive update surrogate mapping between input variables and output objectives is constructed within the surrogate model AOU-1D-CAE framework. AOU-1D-CAE can replace electromagnetic (EM) simulation software for data collection, and select and automatically use the accumulated data as training samples to train the AOU-1D-CAE surrogate model. With more and more training samples, the learning ability of the surrogate model is also becoming stronger and stronger. The experimental results show that the data collection efficiency of AOU-1D-CAE is greatly improved, and the automatic update of the sample set improves the prediction performance of the surrogate model. In this paper, the optimization framework is AOU-1D-CAE-assisted particle swarm optimization (PSO), and the surrogate model assists PSO to find the global optimal solution. In the PSO stage, PSO automatically updates and saves the optimal solution, and takes the optimal solution of each stage as the initial solution of the next optimization stage to avoid falling into the local optimal solution. The optimization time is greatly saved and the optimization efficiency is improved. The continuous iteration of PSO also improves the prediction accuracy of the surrogate model. The efficiency of the proposed surrogate model is demonstrated by using two cavity filters as examples.

**Keywords:** microwave filter design; one-dimensional convolutional autoencoders; online updating; particle swarm optimization; surrogate model

## 1. Introduction

Full-wave electromagnetic (EM) simulation-based optimization technology has become an essential tool for microwave filter design [1,2]. The purpose is to adjust the geometric parameter of the filter to ensure that the frequency response is within design specifications. Full-wave EM simulation is necessary for microwave device optimization [3], and its computational cost is very high. Optimizations on the basis of EM are also very time-consuming, as they usually require repeating EM simulations for various geometrical parameters as design parameters. Traditional EM optimization takes a long time to obtain the best solution to meet the specifications of the design. Therefore, obtaining the best solution in a relatively short period of time is a challenge.

The local optimization approach and the global optimization approach are two types of different methods, which are used in EM optimization, and applied to design and optimize filters. Several local optimization approaches were exploited for EM-based design closures, such as neighbor-sensitivity based on optimization [4,5], and space mapping based on optimization [6,7]. Nevertheless, because the problem is severely multi-patterned [8], local optimizations are inadequate and a global search needs to be performed. Various global

optimization approaches like simulated annealing (SA) [9], particle swarm optimization (PSO) [10], and genetic algorithms (GA) [11] were successfully developed. The algorithms have a random nature which can assist in avoiding falling into a local optimum. The design of filters often has complex and multi-objective aspects. Reaching the optimal variable is related to the objective function under multiple objectives [12,13]. However, the shortcoming is the comparatively slow rate of convergence in these algorithms. In order to achieve a marked reduction in calculation costs, machine learning technologies are used to build surrogate models in place of computing costly EM simulations.

Artificial neural networks (ANN) [14], Kriging [15,16], Support vector machine (SVM) [17], and Gaussian processes (GP) [18] are some of the highly popular machine learning algorithms. They have many variants or combinations [19]. This is an efficient way of getting effective surrogate models. Such algorithms can specifically feature the unit cells and act as a replacement for full-wave EM tools with local regularity. Rapid alternative models are becoming increasingly important in the development and design automation of modern microwave device structures. There are various alternative assisted optimization methods for microwave devices to optimize the problem. The surrogate model assisted evolutionary algorithm (SAEA) is one of the most developed methods. The principle is to use a surrogate model instead of simulations. As the computation of the exact function evaluation is very expensive, the surrogate model uses much less computational overhead than the exact function evaluation of the simulation software. In [15], the forecasts of the various surrogate models are weighted to aid the evolutionary algorithm (EA), and the weights can be adaptively adapted according to the forecasts of the various surrogate models. Cai et al. introduced SAEA to the area of antenna design and synthesis optimization [20]. The surrogate model and the evolutionary algorithm are in a cooperative relationship in SAEA, in contrast with the surrogate model in the spatial mapping-based method. It is worth noting that with the widespread use of ANN in different fields, it is also gaining an interest in the field of microwave design. As a result, many surrogate models currently use ANN in the framework structure.

In recent years, as one of the most well-known machine learning methods [21,22], neural transfer functions (neuro-TF) [23] were developed, which are alternatives to the combination of neural networks and transfer functions and serve to parametrically model and optimize the design of microwave devices [24]. The use of ancillary feature frequencies derived from neural TF [25] is an agency-based model for EM optimization. The principle is to use feature frequencies to help with agent-based model EM optimization in order to avoid getting stuck in a local optimum in the optimization process. However, the traditional ANN model has too many layers, which makes the network more easily over-fitted. In addition to this, EA acts as a search engine in most SAEA, generating new points primarily for optimization, but as the EA generates more new data, there is no guarantee of the accuracy of the surrogate model and thus the feasibility of the EA generating new solutions. A one-dimensional convolutional autoencoders (1D-CAE) based surrogate model was first proposed in [26] to optimize the filter. However, the EM simulation software takes too long to collect training samples, is computationally intensive, and the overall optimization process tends to fall into local optimal solutions.

In this paper, a new approach, adaptive online updating 1D convolutional autoencoders-assisted particle swarm optimization (AOU-1D-CAE-APSO), is introduced which can solve this type of problem. The innovation of AOU-1D-CAE-APSO is the new SAEA framework, adaptive online updating 1D convolutional autoencoders (AOU-1D-CAE), consisting of a less complex ANN. The surrogate model constructs a non-linear relationship between the filter and the geometric parameters. After the training data has been collected by the EM simulation software to make the surrogate model achieve prediction accuracy, the surrogate model can replace the EM simulation software for training data collection. While accelerating the convergence of the optimization algorithm, the network parameters can be adaptively updated online to improve the prediction accuracy of the model embedded in the optimization algorithm.

The sections of this paper are listed below. Section 2 describes the fundamental technologies which make up the AOU-1D-CAE. Section 3 explains the specific work of AOU-1D-CAE-APSO. Section 4 describes two cavity filters using the AOU-1D-CAE-APSO method of optimization experiments and results. Section 5 provides a summarized conclusion.

## 2. Basic Theory

AOU-1D-CAE and PSO are the two key parts of this paper. AOU-1D-CAE is used as the surrogate model and the PSO algorithm is the search machine. A brief description of these techniques follows.

### 2.1. AOU-1D-CAE

Autoencoders (AE) is an excellent method for learning compression and distributed feature representation from a given dataset. The AE comprises an input layer, a hidden layer, and an output layer. A coding process between the input layer and the hidden layer is used to obtain an encoded representation through the coding operation on the input data. Between the hidden layer and the output layer, there is a decoding process, which obtains the reconstructed input data by decoding operations on the hidden layer. Figure 1 shows the basic framework of AE. The encoding process can be expressed as:

$$y = f(Wx + b) \tag{1}$$

where $x$ is the input signal, $y$ is the hidden layer, and $f$ is a nonlinear function. $W$ is the weight matrix. $b$ is the bias matrix. The formula for the decoding process is:

$$\hat{x} = f'(W'y + b') \tag{2}$$

where $\hat{x}$ is the decoded output, which is the actual output of the network, and the predicted value of the input layer $x$, $f'$ is a nonlinear function. $W'$ is the weight matrix. $b'$ is the bias matrix.

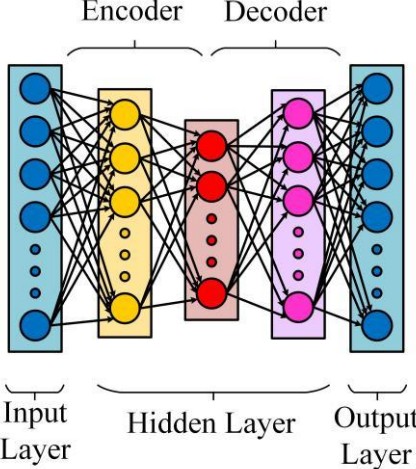

**Figure 1.** AE framework.

In this paper, the inputs to the encoding process are the real and imaginary parts of the *S*-parameters, and the outputs are the geometric parameters. The output of the encoding process is used as the input to the decoding process, and the reconstructed *S*-parameters are used as the output of the decoding process.

The purpose of the AE is to reconstruct the input data, which means that the AE is allowed to learn a function of $x = \hat{x}$. This allows the network to learn different representations of the input data that characterize the data. The AE, in order to reproduce the source input, has to capture the parts that can represent the important features of the source data. The disadvantage of the traditional AE model is the difficulty in extracting features of

the input data. Thus, the AOU-1D-CAE model in this paper combines the 1dimensional convolutional neural network (1D-CNN) with the AE, which means the encoding and decoding process is performed by 1D-CNN. Figure 2 shows the framework of a 1D-CNN.

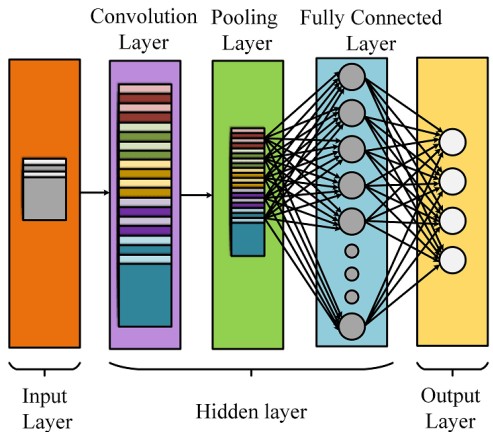

**Figure 2.** General CNN structure.

CNN is a feedforward neural network with a deep network structure. The principle is to extract features from the input data by means of multiple filters. The basic layer types in this paper model are the convolution layer, pooling layer, and fully connected layer. The Convolutional (Conv) layer is the basic structural unit, and the Conv kernel parameters are shared globally, reducing parametric numbers. The number of feature layers can be controlled using the size of the feature map in the Conv layer to learn more and more physical features. The calculation formula of the Conv layers is as follows:

$$y_j^{l+1} = f(\sum_{i=1}^{M} x_j^i \otimes k_{ij}^l + b_l^l) \tag{3}$$

where $y_j^{l+1}$ is the input of j-th neuron at layer *l + 1*; $f(\cdot)$ is activation function; *M* is the feature map; $x_i^l$ which is the output of neuron *j-th* of layer *l*; $\otimes$ means convolution operation; $k_{ij}^l$ is the convolution kernel of *i-th* neurons in layer *l* with *j-th* neurons in layer *l + 1*; $b_i^l$ is the bias.

The pooling layer reduces the amount of data and network overfitting during the solving process. What is used is to extract the maximum value of the pooling window as the output result and then keep sliding the window to reduce the size of the input data, thus reducing redundant information. The calculation process of maximum pooling is as follows:

$$p_j^{l+1}(i) = \max_{(i-1)W+1 \leq t \leq iW} \left\{ a_j^l(t) \right\} \tag{4}$$

where $p_j^{l+1}(i)$ is the corresponding value neurons in layer *l + 1* of the pooling operation; *W* is the width of the pooling region.

A fully connected layer expands the input to the previous layer with a column vector, with an all-connected layer, it acts as a classifier in CNN. The fully connected layer is computed as follows:

$$z_j^{l+1}(i) = \sum_{i=1}^{n} w_{ij}^l p_j^l + b_j^l \tag{5}$$

where $z_j^{l+1}$ is the output of *i-th* neurons of layer *l + 1*; $w_{ij}^l$ is the weights of *i-th* neurons in layer *l* with *j-th* neurons in layer *l + 1*; $P_j^l$ represents the corresponding value of the neuron in layer *l* of the pooling operation; *b* is the bias.

### 2.2. PSO

In the AOU-1D-CAE-APSO algorithm, PSO is used as a search engine. PSO is an evolutionary algorithm that simulates the feeding behavior of a flock of natural species of birds. When searching for food in an area, birds are not sure how far away their current position is from the food, so they need to search the area around the nearest bird to the food. The particle is the simulation of the bird in the algorithm. Each particle has two attributes: velocity and position. Particle *i* has two properties, velocity, and position. At time *g*, the velocity and position of the particle are denoted by the subscripts $v_{i,g}$ and $x_{i,g}$. The velocities show the steps of motion and the positions represent the directions. Each particle finds an individual optimal solution, and the global optimal solution is obtained by comparing the individual optimal solutions found for all particles. Multiple iterations keep the velocity and position updated until the termination condition is met when the loop is exited and the iteration ends.

The velocity and position of each individual (particle) are renewed by the following equation:

$$v_{i,g+1} = v_{i,g} + c_1 \times rand_{i,1} \times (x_{i,pest} - x_{i,g}) + c_2 \times rand_{i,2} \times (x_{gpest} - x_{i,g}) \tag{6}$$

$$x_{i,g+1} = x_{i,g} + v_{i,g+1} \tag{7}$$

$x_{i,g} = (x_1, x_2, \ldots, x_d)$ represents the position of the *i-th* individual in the *d*-dimensional search space at the current PSO iteration to the *g-th* generation, $v_{i,g} = (v_1, v_2, \ldots, v_d)$ represents the speed of the *i-th* individual in the *d*-dimensional search space, $x_{i,pbest}$ represents the historical optimal position of the *i-th* individual in the *d*-dimensional search space, $x_{gbest}$ represents the optimal position of all individuals of the entire population in the *d*-dimensional search space during the optimization process. $c_1$ and $c_2$ represent the learning rates for learning from the *i-th* individual historical optimum and the global optimum, respectively, $rand_{i,1}$ and $rand_{i,2}$ is the random number between [0, 1]. The flow of the PSO algorithm is shown in Figure 3.

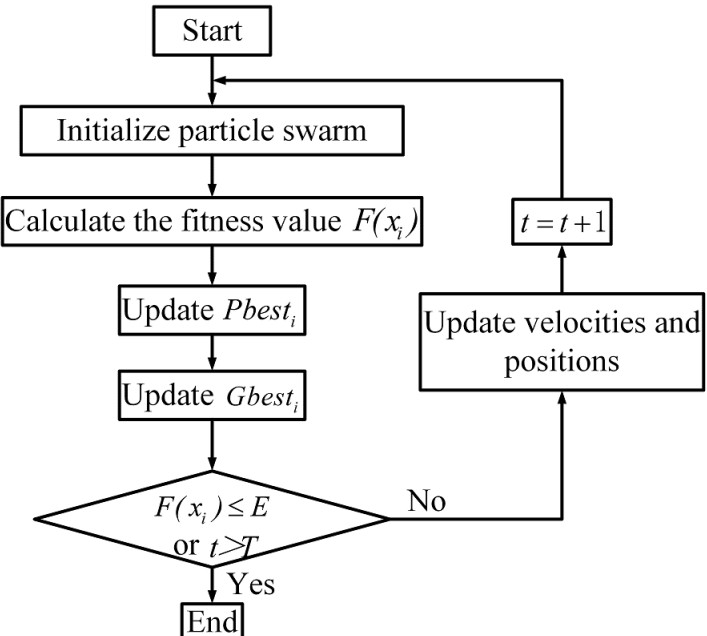

**Figure 3.** The flow of PSO.

## 3. AOU-1D-CAE-APSO Algorithm

### 3.1. Algorithm Framework

The AOU-1D-CAE-APSO framework is shown in Figure 4, which works as follows:

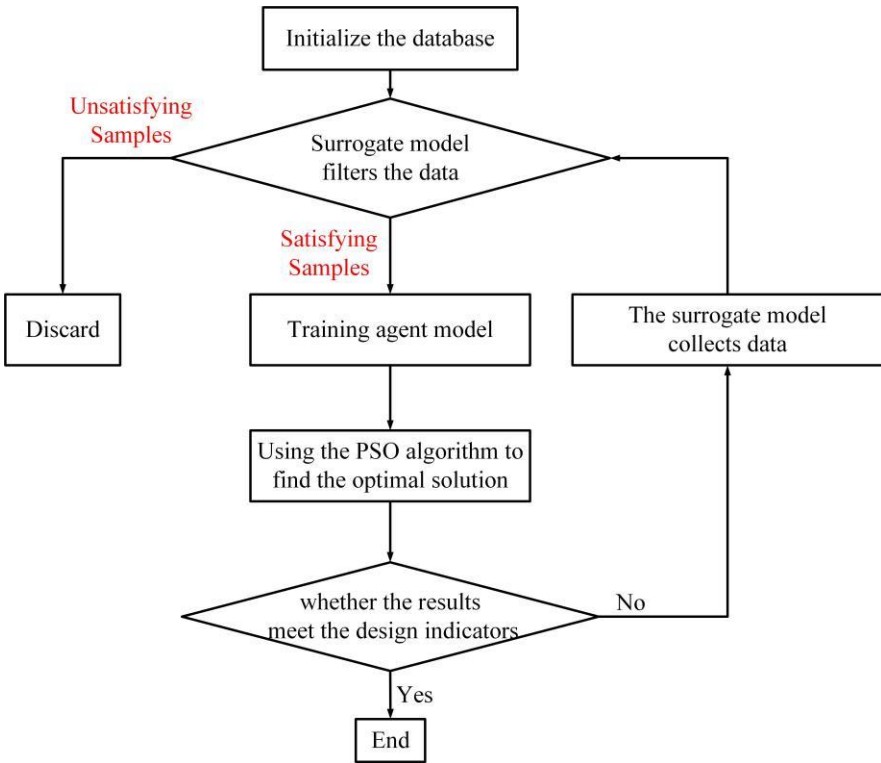

**Figure 4.** Framework of AOU-1D-CAE-APSO.

Step1: Python and High-Frequency Structure Simulator (HFSS) co-simulation to form the initial database;

Step2: Select the samples in the database, and if the preset conditions are met (e.g., the samples in the database meet the satisfaction criteria), then keep the data sample; otherwise delete the sample;

Step3: The database samples are trained to make the surrogate model with certain accuracy and prediction function;

Step4: Add an optimization algorithm that acts as a search engine for the overall optimization process. Embed the surrogate model into the optimization algorithm and use the update iterations of the algorithm to find the optimal solution;

Step5: Determine if the design specifications are met, and if the preset stopping criteria are met (e.g., the reflection coefficient in the optimized pass-band range is reduced to below −20 dB), output the best solution from the optimization program; otherwise go to Step 6;

Step6: The surrogate model captures data to form a second generation database;

Step7: Select the samples in the database, if the preset conditions are met (e.g., the samples in the database meet the satisfaction criteria), then keep the data sample; otherwise delete the sample;

Step8: Combine the initial database samples and the second generation database samples to train the agent model, so that the surrogate model has a certain accuracy and a prediction function;

Step9: The optimization algorithm performs the optimization and comes up with the optimal solution;

Step10: Determine if the design specifications are met, if the preset stopping criteria are met (e.g., the reflection coefficient in the optimized pass-band range is reduced to below −20 dB), then output the best solution from the optimization program; otherwise go to Step 6.

In the data collection stage, the collected real data content is the geometric parameters and their corresponding S parameters. The input of the autoencoder is the S-parameters of the real data, and the input of the PSO is the geometrical parameters of the real data.

The adaptive online update in the surrogate model structure has two innovations: (1) Replaces EM simulation software to reacquire data. A significant saving in filter design time and increased efficiency. (2) When the network prediction is not accurate, the surrogate model automatically collects data near the predicted value and retrains the network to make the network more accurate and improve the prediction performance.

### 3.2. Parameter Settings of AOU-1D-CAE-APSO

There are several parameter settings in the AOU-1D-CAE-APSO, some experimental principles are presented in this paper.

Data Pre-processing. In [27], it is stated that the neural network model accepts the geometry and frequency variables of the filter as input. The values of the geometric and physical frequency variables can change by many orders of magnitude. To address the problem that for *s*-shaped neurons, too large an input can lead to saturation, when the derivative of the *s*-shaped function is very close to zero, slowing down the learning process of the ANN model, data pre-processing is required. Within the range of values of the input variable $x$ [$x_{min}$, $x_{max}$], it can be linearly scaled to [$-1,1$]:

$$\overline{x} = 2\frac{x - x_{min}}{x_{max} - x_{min}} - 1 \tag{8}$$

Equation (8) shows the linear mapping between the physical value $x$ and the ANN model input value $\overline{x}$. The mapping calculates the physical value $x$ to obtain the optimized neural network model input value $x$:

$$x = \frac{(x_{max} - x_{min})}{2}(\overline{x} + 1) + x_{min} \tag{9}$$

Parameter setting in the surrogate model. The number of training data is decided by the trade-off between the surrogate model quality and the surrogate model training time. In this paper, the optimization process is intelligent. When the accuracy of the surrogate model is not high enough, training sample data can be added automatically to train the surrogate model. The detailed internal structure of the surrogate model is shown in Figure 5, including the layers, size/Nodes, Stride, and activation functions for each layer. The network structure of the surrogate model is shown in Figure 6.

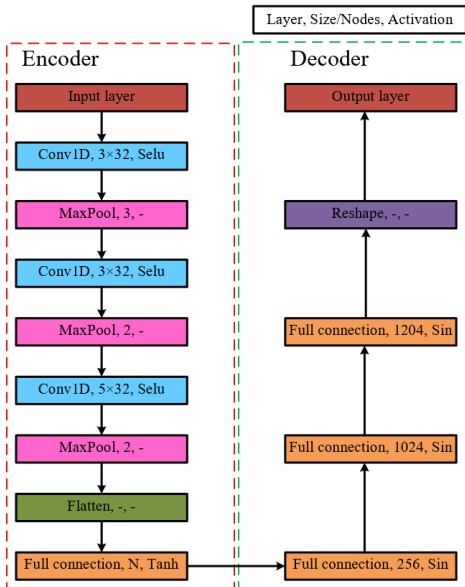

**Figure 5.** Detailed architecture of 1D−CAE in AOU−1D−CAE−APSO.

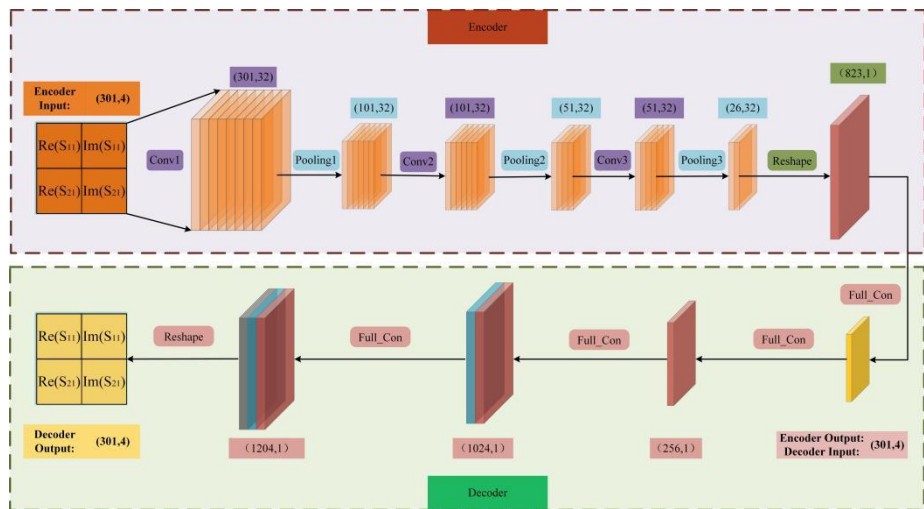

**Figure 6.** Network structure of the AOU-1D-CAE.

The loss function of the surrogate model is cited from the literature [26]: a combination of two loss functions, reconstruction loss ($L_r$) and prediction loss ($L_p$). The combined loss function is used to train the entire network. In the AOU-1D-CAE surrogate model, the real and imaginary parts of the input data $S_{11}$ and $S_{21}$ are encoded as geometric parameters. The decoder reconstructs the real and imaginary parts of $S_{11}$ and $S_{21}$. Reconstruction losses were used to evaluate the reconstruction performance of the AOU-1D-CAE models. The reconstruction loss is expressed as follows:

$$L_r = \frac{1}{N}\sum_{i=1}^{N}\left|S_t - S_p\right| \tag{10}$$

where $N$ is batch size. $S_t$ is the simulated $S$-parameters, $S_p$ is the reconstructed $S$-parameters of the 1D-CAE model. The prediction loss is defined as the mean square error between the predicted value of the encoder and the actual value of the geometric parameters:

$$L_p = \frac{1}{N}\sum_{i=1}^{N}\left|H_t - H_p\right| \tag{11}$$

where $N$ is batch size. $H_t$ is the actual value of the geometric parameters, $H_p$ is the prediction value of the encoder. Finally, the overall loss function is the sum of $L_r$ and $L_p$, which is defined as follows:

$$L = L_r + \lambda L_p \tag{12}$$

where $\lambda$ is a regularization parameter. The loss function of this network is optimized by using the Adam optimizer.

The parameters in the search framework: in PSO, the position of the particles is the geometric parameter to be optimized in the filter. The search dimension is the number of geometric parameters. The fitness function, cost, takes points evenly across the range of the pass-band to be optimized by $S_{11}$, takes the distance between each point and the target value, and sums these values. PSO is responsible for searching for the position of the particle corresponding to the minimum value of the cost function.

## 4. Experiment and Results

### 4.1. Fourth-Order Cavity Filter

In the first example, consider the optimization of a fourth-order cavity filter [28] to demonstrate the proposed optimization technique. Figure 7 illustrates the structure of a fourth-order cavity filter. The fourth-order cavity filter is centered at 2.0693 GHz and the

bandwidth is 110 MHz. The design specifications are defined as $|S_{11}| \leq -20$ dB in the frequency range from 2.0143 GHz to 2.1243 GHz. Each EM simulation costs about 3–4 min on average. The fourth-order cavity filter needed to optimize the geometric parameters are $X = [p_1, p_2, p_3, p_4]^T$. The optimization objective is the minimization of the fitness function, cost in Equation (13), to satisfy the design specifications. In PSO, the fitness function is:

$$Cost = \sum_{n=1}^{6} d_n - (-20) \times 6 \tag{13}$$

where $d_i$ is the distance between the six points and the optimization target within the optimization pass-band on $S_{11}$, after taking six points at equal intervals. *Cost* is the fitness function of PSO, which seeks to find the minimum. The internal process of the AOU-1D-CAE-APSO is shown in Figure 8.

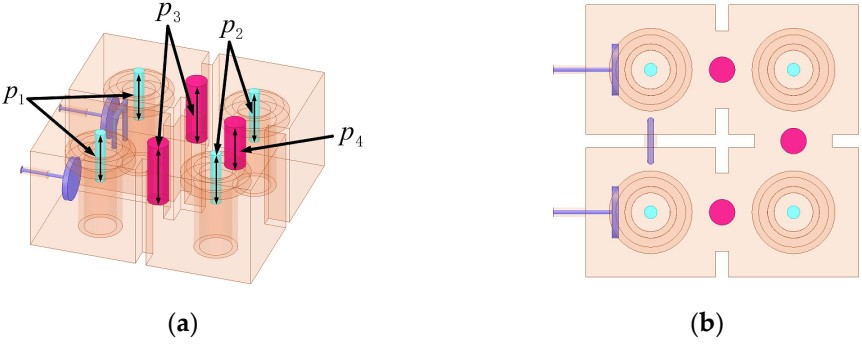

(**a**)  (**b**)

**Figure 7.** Fourth-order cavity filter. (**a**) Side view. (**b**) Top view.

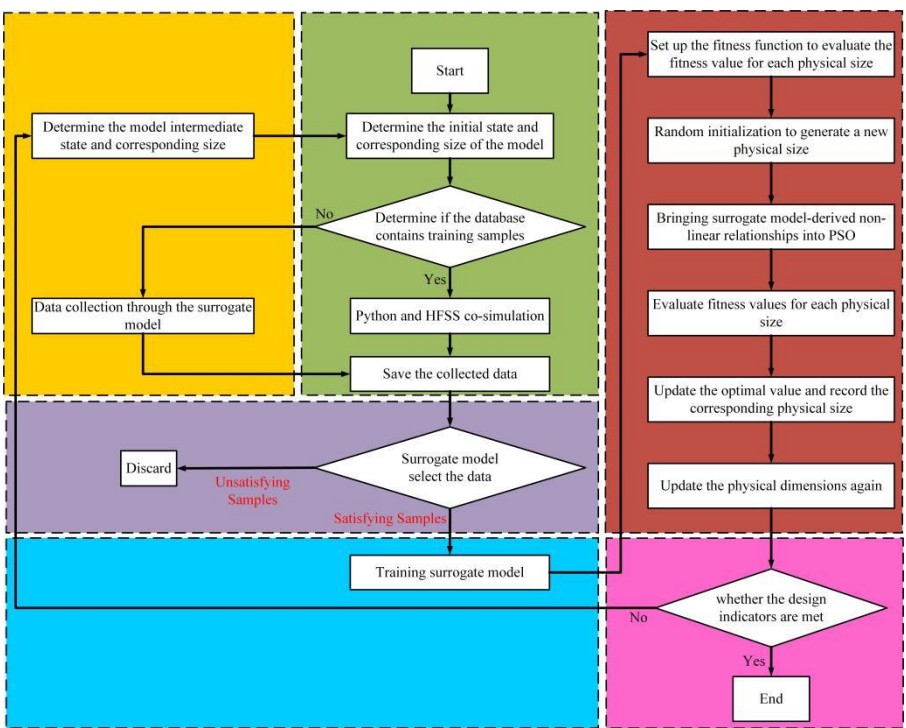

**Figure 8.** Internal process of the AOU-1D-CAE-APSO.

During the design of the microwave filter, the initial state of the filter is not ideal, use this stage as stage 1. The $S_{11}$ of the filter before optimization is $-5.3$ dB, corresponding to the geometric parameters taken as $p_1 = p_2 = p_3 = p_4 = 10$ mm. The response results of the

PSO algorithm iteration are shown in Figure 9. The initial state is shown as the first stage in Figure 9.

**Figure 9.** Fourth−order cavity filter HFSS verification of the optimized response in PSO iterations. (**a**–**e**) Reflection coefficient $S_{11}$. (**f**) Transmission coefficient $S_{21}$.

Firstly, add training samples. This paper proposes an optimized design method that only uses a co-simulation of python and HFSS for the first data collection. More data is obtained by disturbing the geometrical parameters, as in Equation (14),

$$p_n' = p_n \pm p_n \times 0.5\% \ (n = 1, 2, 3, 4) \tag{14}$$

the new geometrical parameters obtained are input to HFSS for simulation. For example, $p_1 = 10$ mm, $p_1' = p_1 \pm p_1 \times 0.5\%$, get a new set of geometric parameters $H' = [p_1', p_2, p_3, p_4]^T$, the remaining geometric parameters get new data according to this disturbing index, consist of new geometric parameters, then input into HFSS for simulation, get the sample data. The HFSS outputs are $Re(S_{11})$, $Im(S_{11})$, $Re(S_{21})$, $Im(S_{21})$. The sweep range is 1.85–2.2886 GHz, and each curve consists of 301 points. The resulting data are saved as npy files, the content of each npy file containing the geometric parameters, and the corresponding four *S*-parameter curves. The first data collection in this experiment was obtained from HFSS simulations and 300 data samples were collected over 930 min. Save as npy file. The samples are classified into 240 training data sets and 60 test data sets. Approximately 20 pieces of data were collected on average per hour.

Next step, select data. A selection range is usually defined. In this experiment, the initial state $S_{11}$ is −5.3 dB, and the pass-band range is from 2.0143 to 2.1243 GHz. The optimization target is $S_{11} \leq -20$ dB. Set the selection range to [−10 dB, −5.3 dB], the data not in this range will be deleted, and the satisfactory data will be kept. 297 of these files met the conditions. 3 files were deleted.

Then start building the AOU-1D-CAE surrogate model. AOU-1D-CAE is a combination of AE and CNN. Generally divided into two parts, encoder and decoder. Train the surrogate model. The collected data samples were put into the surrogate model and the neural network was trained. $Re(S_{11})$, $Im(S_{11})$, $Re(S_{21})$, $Im(S_{21})$ from a real sample is used as input to the surrogate model, with each *S*-parameter consisting of 301 points. First, an encoding operation is performed. In the encoder, the input matrix is $301 \times 4$. After three mutually alternating convolution and pooling layers, the matrix becomes $26 \times 32$. After

flattening the matrix becomes a vector, and then after a full-connected layer, prediction is performed. The output of the encoder is the corresponding geometric parameter obtained by the prediction of the $S$-parameter. The output of the encoder is then used as the input to the decoder for the decoding operation. After three fully connected layers in the decoder, the network is then reshaped to a matrix $301 \times 4$. The loss function of this network is Equation (12). The neural network learns the non-linear relationship between the geometrical parameters and the corresponding $S$-parameters and is able to predict the physical performance corresponding to any set of geometrical parameters with a loss function that meets the criteria. After building the surrogate model, the surrogate model was trained on NVIDIA Geforce GTX 1060 6 GB using the GPU environment. Experiments show that it takes 7 min to train the network using the GPU environment and 27 min using the CPU environment. Each time the surrogate model is trained, the prediction accuracy of the network is checked. A random set of geometric parameters $X$ is chosen to be placed into the AOU-1D-CAE and the network outputs a set of $S$-parameters. Figure 10a illustrates the training and testing performance of the network. It can be seen that the model has a faster convergence rate and higher accuracy as the number of training sessions increases. Figure 10b presents a contrast between the HFSS simulation results and the predicted results from the AOU-1D-CAE model. It can be seen from Figure 10 that the results match exactly.

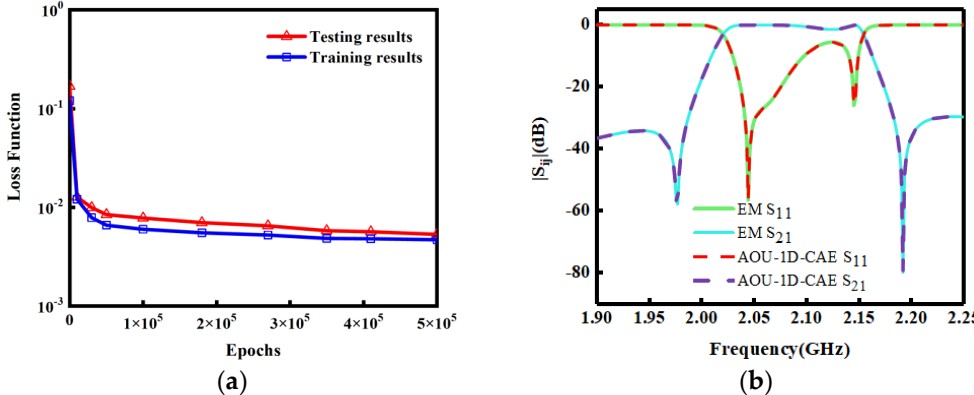

**Figure 10.** (**a**) AOU−1D−CAE neural network training and testing result. (**b**) Fourth−order cavity filter HFSS simulation results with surrogate model network prediction results.

The PSO were implemented using the open source and well-developed Python codes. The fitness function is the key to the PSO algorithm's important basis for evaluating whether the computed solution satisfies the design objective. The bandwidth to be optimized is intercepted, that is, $S_{11}$ whose frequency range is between 2.0143 GHz and 2.1243 GHz is intercepted. Take 6 points ($d_1, d_2, d_3, d_4, d_5, d_6$) at equal intervals on the intercepted $S_{11}$, take the distance between these 6 points and $y = -20$ dB, and the sum is the fitness function of PSO. The fitness function is Equation (13), and the goal is to find the minimum value.

In this experiment, the particle dimension is 4 (the number of geometric parameters is 4). The number of particles is 100. The maximum number of iterations is 100. The acceleration factor $c_1 = c_2 = 2$. Once initialization is complete, a new set of geometric parameters is generated at random, bringing surrogate model-derived non-linear relationships into PSO. The resulting geometric parameters are fed into the surrogate model and the $S$-parameters are obtained by prediction. Calculate the fitness function for each particle. Update $x_{gbest}$ and $x_{i,pbest}$.

The position of the $x_{gbest}$ is at this point the optimal size parameter for the filter. After PSO iterations, the optimal particle in the population is derived as follows: $P_1 = 9.7275$ mm, $P_2 = 10.1420$ mm, $P_3 = 9.6737$ mm, $P_4 = 9.6804$ mm. The corresponding values are put into the surrogate model for prediction and a set of $-9.8$ dB reflection coefficients is obtained. Determine if the responses meet the design specifications. If the result does not meet, go to the next step.



As shown in Figure 9b, the model has been optimized from stage 1 to stage 2, with a decrease in $S_{11}$ in the optimized pass-band. The reflection coefficient corresponding to stage 2 is $-9.8$ dB. At this point, the geometric parameters corresponding to stage 2 are used as the initial state and the training sample data are collected. The second data acquisition was performed using a surrogate model instead of HFSS simulation software. AOU-1D-CAE replaces HFSS for data acquisition, the same perturbation is carried out using Equation (14) to obtain more new geometric parameters. The new geometric parameters are put into the surrogate model as input and predicted by the surrogate model to produce *S*-parameters from which new training sample data can be composed. The second data collection in this experiment was obtained from AOU-1D-CAE simulations and 300 data samples were collected over 1 min. Save as npy file. The contents of the npy file are the same as the first time training data was collected.

Data samples in the range $[-15, -9.8]$ dB are selected, and data not in this range are deleted. The AOU-1D-CAE obtained 300 npy files and took 1 min. 297 of these files met the conditions. 3 files were deleted.

The 297 data samples from the first collection were fused with the 297 data samples from the second collection and put into the surrogate model for training.

Bringing surrogate model-derived non-linear relationships into PSO. After PSO iterations, the optimal particle in the population is derived.

The position corresponding to the optimal particle, the geometric parameter *H*, is put into the surrogate model for prediction, and a set of reflection coefficients is obtained. At this point the reflection coefficient is 13 dB, corresponding to stage 3 of Figure 9c, which does not meet the design specification. Confirming this set of geometric parameters *H* as the initial state, the data around this set *H* is collected using the surrogate model with reference to the previous perturbation indicators.

Repeat the optimization process, meeting the design specifications when proceeding to stage 5. Exit the cycle. Figure 11 shows the details of each stage of the AOU-1D-CAE.

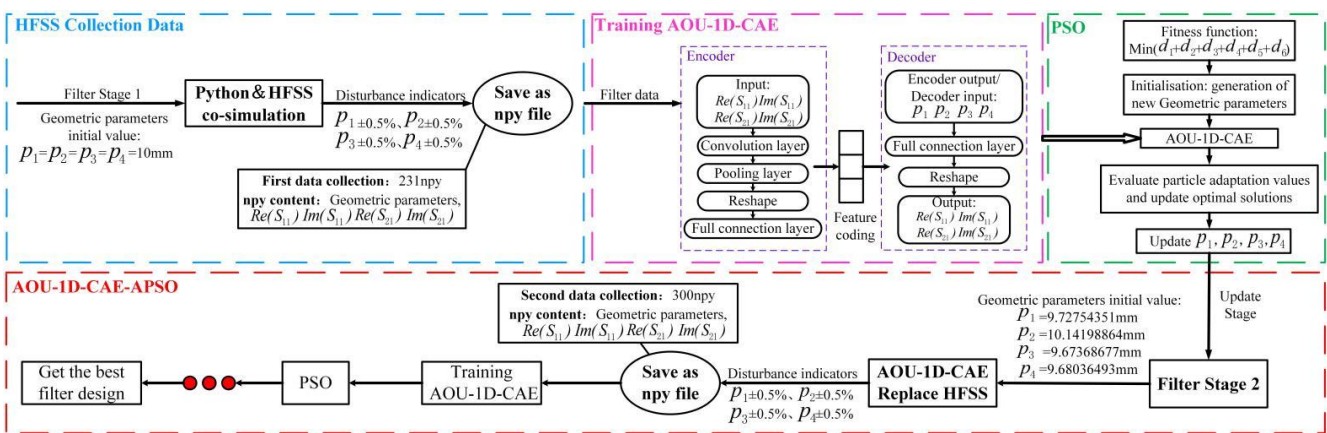

**Figure 11.** Details of the AOU-1D-CAE optimization.

AOU-1D-CAE-APSO obtains greatly increased speed improvement compared with standard full-wave simulation for this example, making the unbearable time to be very practical and obtaining an even better result. As shown in Table 1, comparing the two ways of collecting training data samples, some observations can be made. Optimizing filters with traditional full-wave simulations can take hundreds of minutes to calculate, whereas the optimization method proposed in this paper reduces the time greatly, and the surrogate model is able to collect 300 sets of data samples in 1 min, compared to 930 min for HFSS. The comparison of time, as well as the computational cost, is very obvious. Table 2 gives a comparison of the time taken to optimize the filter by the traditional optimization approach [26] and by the two surrogate model-assisted PSOs.

**Table 1.** Fourth-order cavity filter optimization process.

| Stage | Initial State (dB) | Data Collection | Time | Satisfaction Data | Reflection Coefficient Distributed Area (dB) |
|---|---|---|---|---|---|
| 1 | −5.3 | 300 | 930 min | 297 | [−9.8, −5.3] |
| 2 | −9.8 | 300 | 1 min | 297 + 297 | [−13, −9.8] |
| 3 | −13 | 300 | 1 min | 295 + 297 + 297 | [−16.2, −13] |
| 4 | −16.2 | 300 | 1 min | 286 + 295 + 297 + 297 | [−20, −16.2] |
| 5 | −20 | − | − | − | − |

**Table 2.** Two surrogate model-assisted particle swarm optimization algorithms.

| Step | Time | |
|---|---|---|
| | Traditional SEAE | AOU-1D-CAE-APSO |
| Initialize the database | 930 min | 930 min |
| Train the surrogate model | 27 min | 7 min |
| PSO | 1 min | 1 min |
| Adding training samples | 930 min | 1 min |

*4.2. Eighth-Order Cavity Filter*

The first experiment optimized four parameters of the filter, this experiment will optimize eight parameters and prove whether the algorithm works.

In this section, experiments were carried out with an eighth-order cavity filter [29], the model of which is illustrated in Figure 12. The metrics are as follows, with a center frequency of 2.0693 GHz and a bandwidth of 110 MHz, and the geometrical parameters to be optimized are $H = [p_1, p_2, p_3, p_4, p_5, p_6, p_7, p_8]^T$. Each EM simulation costs about 12–15 min on average. The design specification for this example is defined as $|S_{11}| \leq -20$ dB in the frequency range of 2.0143–2.1243 GHz. The first time data was collected by HFSS, 485 data were collected in a total of 52.13 h. The data were collected simultaneously by two servers. The optimized pass-band range is 2.0143–2.1243 GHz. The experimental procedure was the same as in Section 4.1. The same parameters and fitness functions are used in PSO. The optimization objective is the minimization of the fitness function, cost in Equation (13), to satisfy the design specifications. Figure 13 presents a contrast between the HFSS simulation results and the predicted results from the AOU-1D-CAE model. Figure 14 illustrates the optimization process, the optimization process goes through four cycles and five stages. The $S_{11}$ change from stage 1 to stage 5 corresponds to Figure 14a–e. Figure 14f shows the variation of $S_{21}$. The value of stage five approximates meeting the design requirements. After four cycles, PSO finds the optimal solution. After EM simulation, the design purpose is approximation satisfied. Stage 5 as the final result of the reflection coefficient, differs from the ideal result by −1.4 dB, which is within the acceptable margin of error. Table 2 shows the eighth-order cavity filter optimization process and the data collected. HFSS took 52 h to collect around four hundred data samples, whereas the agent model can give thousands of training data samples in just 4–6 min, so it can be seen that AOU-1D-CAE-APSO saves a lot of computational and time costs.

This experiment illustrates the feasibility of the proposed algorithm. The content of Table 3 is the eighth-order cavity filter optimization process. Table 3 reflects that the advantages of this algorithm for efficient and fast filter design are more evident when more parameters are optimized.

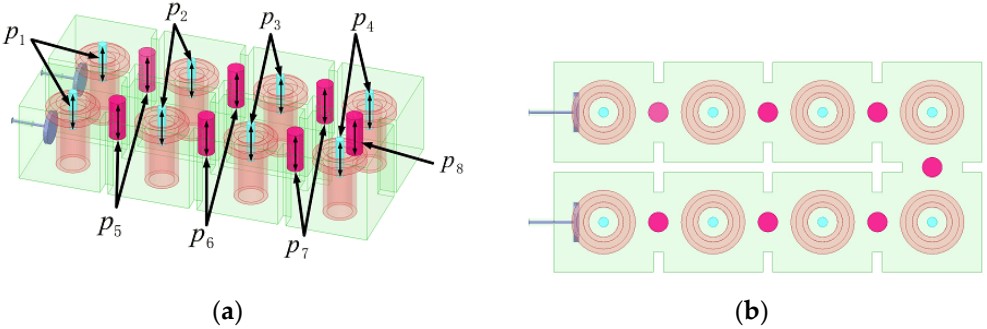

**Figure 12.** Eighth-order cavity filter. (**a**) Side view. (**b**) Top view.

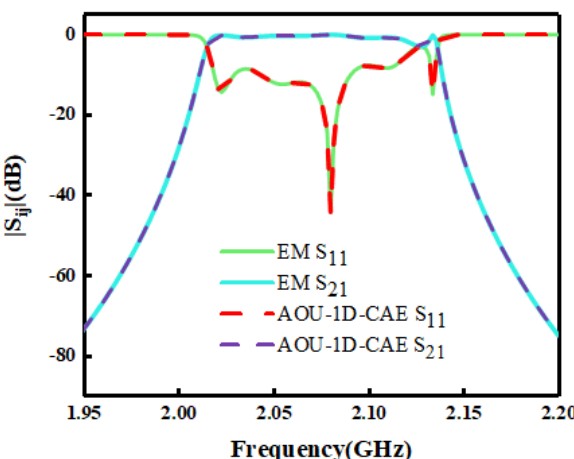

**Figure 13.** Eighth−order cavity filter HFSS simulation results with proxy model network prediction results.

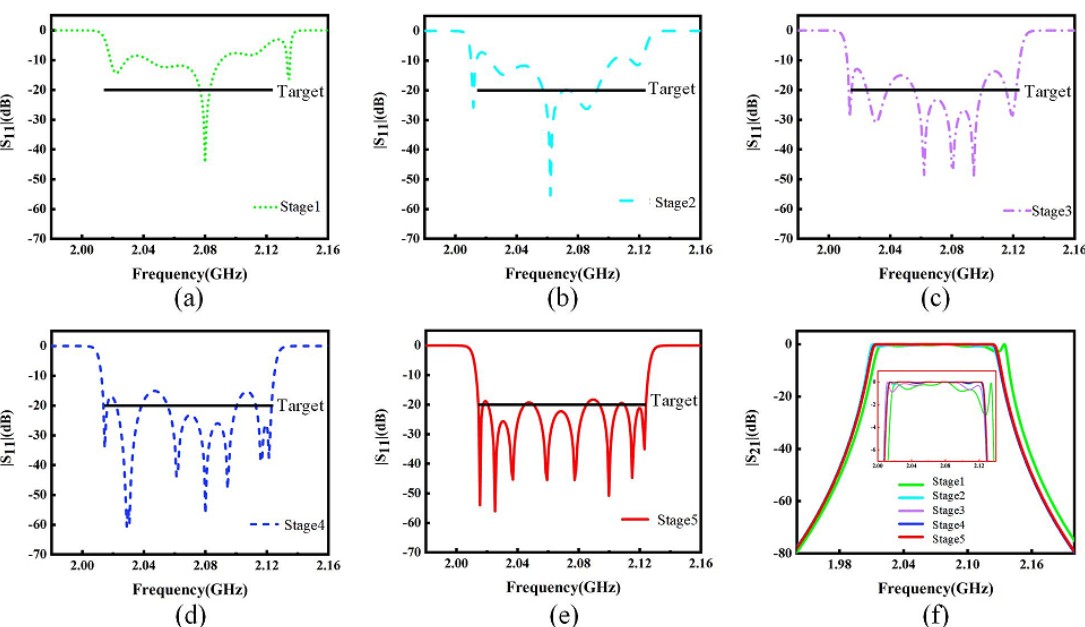

**Figure 14.** Eighth−order cavity filter HFSS verification of the optimized response in PSO iterations. (**a**–**e**) Reflection coefficient $S_{11}$. (**f**) Transmission coefficient $S_{21}$.

**Table 3.** Eighth-order cavity filter optimization process.

| Stage | Initial State (dB) | Data Collection | Time | Satisfaction Data | Reflection Coefficient Distributed Area (dB) |
|-------|--------------------|-----------------|------|-------------------|-----------------------------------------------|
| 1 | −7.3 | 485 | 52.13 h | 469 | [−10, −7.3] |
| 2 | −10 | 1600 | 4 min | 1577 + 469 | [−15.4, −10] |
| 3 | −15.4 | 1600 | 4 min | 1563 + 1577 + 469 | [−17, −15.4] |
| 4 | −17 | 3200 | 6 min | 142 + 1563 + 1577 + 469 | [−18.6, −17] |
| 5 | −18.6 | — | — | — | — |

## 5. Conclusions

This paper presents a new microwave filter design approach, AOU-1D-CAE-APSO. The AOU-1D-CAE-APSO model is simple in structure and low in complexity. AOU-1D-CAE can adaptively update the content and network parameters of the dataset. Replacing EM simulation software for data collection reduces computational costs and training time. The selection function is added to automatically delete the data that is not in the collection range. At the same time, PSO performs phased optimization and updates the phase optimal solution in time. By inputting the continuously updated stage optimal solution into the PSO, the final optimal solution is obtained through optimization. The final optimal solution refers to the solution that finally meets the design criteria. Avoid falling into the local optimum and ensure the feasibility of the solution.

**Author Contributions:** Conceptualization, Y.Z., X.W. and Y.W.; methodology, Y.Z., X.W. and Y.W.; software, Y.Z., X.W., Y.W. and N.Y.; validation, Y.Z., X.W. and Y.W.; formal analysis, Y.Z., X.W. and Y.W.; investigation, Y.Z., X.W. and Y.W.; resources, Y.Z., X.W. and Y.W.; data curation, Y.Z., X.W. and Y.W.; writing—original draft preparation, Y.Z., X.W. and Y.W.; writing—review and editing, Y.Z., X.W., Y.W., L.F. and L.Z.; visualization, Y.Z., X.W. and Y.W.; supervision, Y.Z., X.W., Y.W., L.F. and L.Z.; project administration, Y.Z., X.W. and Y.W.; funding acquisition, Y.Z., X.W. and Y.W. All authors have read and agreed to the published version of the manuscript.

**Funding:** This work was funded by the National Natural Science Foundation of China (NSFC) under Project No.61761032, No.62161032, Nature science foundation of Inner Mongolia under Contract No. 2019MS06006. This work was funded by the Inner Mongolia Foundation 2020MS05059 and Inner Mongolia Department of Transportation NJ-2017-8. This work was also supported by the Shaanxi Key Laboratory of Deep Space Exploration Intelligent Information Technology under Grant No. 2021SYS-04. This work was also supported by the Research and Development of New Energy Vehicle Product Testing Conditions in China—Hohhot Baotou, 2017.

**Conflicts of Interest:** The authors declare no conflict of interest.

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
