# Peer review of "Accurate Design of Microwave Filter Based on Surrogate Model-Assisted Evolutionary Algorithm"

_electronics, doi:10.3390/electronics11223705_

Round 1

Reviewer 1 Report

The PSO algorithm optimizes the geometric parameters to minimize the cost function. The optimization result gives the best position. The autoencoder takes the parameter S as input. However, the relationship between the geometric parameters of PSO and the input of the autoencoder is not clearly presented in the article.

In some parts of the paper, academic and formal language is not used. It is suggested to revise the sentences for easier understanding of the reader.

The authors often use the terms "return loss" for S11 and "insertion loss" for S21. However, this is an incorrect usage. The terms reflection coefficient, scattering parameters or s-parameter should be used for S11 and transmission coefficient for S21. Because return loss is always positive and can never take negative values. For detailed information on this topic, please refer to the publication "Definition and Misuse of Return Loss [Report of the Transactions Editor-in-Chief]" (https://doi.org/10.1109/MAP.2009.5162049).

The sentence in line 172 is written twice in a row.

Reviewer 2 Report

  1. Dear Authors,

Please consider the following comments before resubmission: 

  1. 1. There are multiple linguistic errors in the manuscript that need to be corrected. The first one appears in the title: 

  1. Surrogate Model Assited Evolutionary Algorithm -> Surrogate Model-Assisted Evolutionary Algorithm 

  1. “The principle is to use a surrogate model to instead of the simulations.” -> The principle is to use a surrogate model instead of simulations. 

  1. 2. In the sentence “…. are weighted to aid the evolutionary algorithm (EA), and the weights can be adaptively adapted according to the forecasts of the various surrogate models. the weights are adaptively updated not adapted.  

  1. 3. In the introduction section: you have a sentence: “[20] introduces SAEA to the area of antenna design optimization (synthesis). This is not a proper way to introduce someone else’s work. The typical way is to say the Author’s name, or first author’s name et.al., (if there are more than two authors), or the names of both authors in case of two authors. In your case, it should be “Cai et al., introduced SAEA to the area of antenna design and synthesis optimization [20].  

  1. 4. “Section 5 makes summary.” -> Section 5 derives conclusions from the results. Or Section 5 provides a summarized conclusion.  

  1. 5. The subsection "2.1. Subsection" needs a suitable title or should be integrated into its main section.  

  1. 6. In figure 1., The AE framework is beautifully illustrated. My question is, “Is the Input layer a part of the Encoder and the Output layer a part of the decoder?” Please check and adjust the figure if necessary. I assume, since encoding-decoding is a part of the hidden layer, the input and output layers are not a part of it.  

  1. 7. Your suggested technique, the AOU-1D-CAE-APSO, saves a lot of computational and time costs. This is an interesting outcome but in the abstract section, there is not a clear enough statement to show what are the added features to 1D-CAE. You need to modify the Abstract and Conclusion sections to reflect your main improvement compared to the reference [26]. (Y. L. Zhang, Y. X. Wang, Y. X.Yi, J. L.Wang, J. L, and Z. X. Chen, "Coupling Matrix Extraction of Microwave 561 Filters by Using One-Dimensional Convolutional Autoencoders," Front. Phys., vol. 9, 2021, Art. no. 716881.). So, for example, is “the adaptive update surrogate mapping” a new feature suggested by your team that has improved the previous techniques? Is the PSO (Particle Swarm Optimization) a new feature added to 1D-CAE that improved your results compared to previously published techniques? If yes why not add it to your abbreviated technique name in the abstract and possibly throughout the manuscript?

  1. 8. Improve the English grammar in the whole paper, especially in the Conclusion section. Avoid long sentences and divide them into medium size sentences for clarity. For example, you have a 4-line-long sentence in the Conclusion section.
